# From Ground to Grain: Tracing Phosphorus and Potassium in Flooded Rice Cultivar Grown on Histosols

Naba R. Amgain [1], Yuchuan Fan [1], Matthew T. VanWeelden [2], Abul Rabbany [3] and Jehangir H. Bhadha [1,*]

1   Department of Soil, Water, and Ecosystem Sciences, University of Florida, Everglades Research and Education Center, Belle Glade, FL 33430, USA
2   IFAS Extension, University of Florida, Everglades Research and Education Center, Belle Glade, FL 33430, USA
3   Everglade Research and Education Center, University of Florida, Belle Glade, FL 33430, USA
*   Correspondence: jango@ufl.edu

**Abstract:** To trace the phosphorus (P) and potassium (K) content in flooded rice (*Oryza sativa* L), 14 rice cultivars commonly grown in the Southern United States were evaluated for their P and K concentration in tissue and grain. Field experiments were conducted at two locations in Everglades Agriculture Area (EAA), where flooded rice was cultivated on organic Histosols. Soil pH and Mehlich-3 phosphorus (M3P) were significantly different between locations. At Site I, soil pH, M3P, and Mehlich-3 potassium (M3K) varied in the range of 6.8–7.1, 21.4–36.4 mg kg$^{-1}$, and 53.9–151.0 mg kg$^{-1}$, respectively. At Site II, soil pH, M3P and M3K varied in the range of 6.9–7.3, 11.2–20.5 mg kg$^{-1}$, and 64.8–104.1 mg kg$^{-1}$, respectively. Stem potassium was the only measured parameter that was significantly different among rice cultivars at both sites. At Site I and Site II, stem K ranged from 14.2–26.6 mg kg$^{-1}$ and 10.4–19.4 mg kg$^{-1}$, respectively. No significant difference in yield among cultivars was observed at Site I, whereas Site II had a significant difference in yield among cultivars. At Site I and Site II, yields ranged from 3745–7587 kg ha$^{-1}$ and 2627–6406 kg ha$^{-1}$, respectively. None of the cultivars ranked consistently in the same top and bottom position for each measured parameter. Total phosphorus (TP) concentration was highest in grain, whereas total potassium (TK) concentration was highest in the stem. Results suggest incorporation of rice stem into the soil could potentially add fertilizer back to the soil which helps in fertility management.

**Keywords:** histosol; rice; phosphorus; potassium; plant uptake; cultivar

## 1. Introduction

Phosphorus (P) and potassium (K) are two nutrients essential for plant growth. Deficiency in either of these elements can severely limit crop yields. Phosphorus in the form of sugar phosphate, nucleic acids, and phospholipids is important during plant growth because of its role in the transfer of energy, protein metabolism, root development, and crop yield improvement [1,2]. It is used in the formation of DNA, cell division, and the growth of new tissues [3]. The form of P that is absorbed is inorganic orthophosphate (Pi, $HPO_4^{2-}$, $H_2PO_4^{-}$) [4]. The absorption of Pi is aided by a variety of phosphate ion transport mechanisms [3]. Pi:2H$^+$ cotransporters are responsible for direct absorption from the soil [3]. Phosphate transporters are important for inorganic phosphate absorption from the soil [5]. Potassium plays an important role in enzyme activation, providing electroneutrality during $NO_3^-$ transport, water homeostasis, cell turgor and movement of cells and organs in plants, cell expansion and plant development, and improving plant size, shape, color, and taste [3,6]. Plants deficient in K have reduced turgor pressure, resulting in lower growth rates [6], and reductions in crop yield and plant health [7]. Potassium is absorbed in the ionic form (K$^+$) [4]. The H$^+$ -ATPase drives the uptake of K ions from the soil, which occurs against the K$^+$ concentration gradient [3]. In a sustainable food production system, if these nutrients are removed during harvest, they must be replenished as fertilizer to prevent nutrient depletion and soil degradation.

Growers are finding it difficult to source affordable fertilizer for crop production due to recent geopolitical instability and supply chain challenges associated with fertilizer. The price of fertilizer has increased by 30% in the first quarter of 2022 after jumping 80% the year before. There is little expectation that fertilizer prices will fall in the near term [8]. Therefore, it is beneficial to identify crop cultivars that perform well under little or no fertilizer application. It has been estimated that about 50% of agricultural soils are deficient in P [9], either because of insufficient P being replaced in the systems, or P-fixing soil characteristics rendering the nutrient unavailable to plants (as is the case of the histosols discussed in this paper). In farming regions with P-fixing soils, high fertilizer application is necessary to provide sufficient plant-available P. P-fixing soils are generally soils with low or high pH that form complexes of P with aluminum (Al) and iron (Fe) in acidic soils, or with calcium (Ca) and magnesium (Mg) in alkaline soils [10].

Within the Everglades Agricultural Area (EAA) of South Florida, approximately 10,000 ha of flooded rice is grown every summer on highly organic histosols commonly referred to as "muck" soils. These soils have a soil pH of $7.6 \pm 0.4$, organic matter content $74 \pm 12\%$, total P concentration $1774 \pm 1100$ mg kg$^{-1}$, Mehlich-3 P concentration $123 \pm 170$ mg kg$^{-1}$, and Mehlich-3 K concentration $270 \pm 225$ mg kg$^{-1}$ [11]. The net value of growing rice in the EAA as a rotation crop far exceeds its monetary return. In addition to being a food crop in Florida, the production of flooded rice provides several benefits to the agroecosystem. By flooding fields, growers greatly reduce the negative impacts from issues related to soil subsidence, nutrient depletion, and insect pests. This, in turn, enhances the subsequent sugarcane crop and maximizes the longevity of the soil by reducing soil loss due to oxidation. Incorporating rice as a rotation crop in the EAA during the summer also provides local employment [12]. With no added P or K in the form of fertilizer being applied during rice cultivation in the EAA, this study provides an insight into the behavior of rice uptake of P and K from the soils to the gain. The baseline data from this study will also be used to identify high-yielding cultivars for histosols with reduced nutrient inputs. The objective of this study was to (i) analyze and compare P and K content in 14 rice cultivars grown on flooded histosols; and (ii) identify rice cultivars that have the potential to perform well on histosols without added fertilizer inputs.

## 2. Materials and Methods

This experiment consists of 14 rice cultivars commonly grown in the Southern United States (Table 1). The rice was dry-seeded at 100 kg ha$^{-1}$ in 1.2 m $\times$ 6 m plots with 0.3 m buffer on both edges. The experiment was arranged in a completely randomized design at two locations. Site I (King Sugar) was planted early in the season (24 March 2017), and Site II (Okeelanta) was planted a month later (25 April 2017). Soils at both locations comprised organic histosols, with the soil having a pH of 7.8 and up to 80% organic matter. The weather data is presented in Table 2.

Soil samples at a depth of 0–15 cm (topsoil), and 10–15 mature fully ripe rice plants were uprooted just before harvest from each plot. Both soils and plant samples were air dried at 50 °C for five days. The remaining rice plant from each plot was harvested separately, and rice yields were reported as kg ha$^{-1}$. Dried soils were sieved through 2 mm mesh and stored at room temperature. Rice plants were separated into roots, stems, panicle, and grain (de-husked), ground into powder through a commercial mill, and stored at room temperature. Soil samples were analyzed for pH, TP, TK, M3P, and M3K. The pH of the soil was determined by mixing 1.5 g dry soil with 7.5 mL of deionized water and analyzing the results using an Accumet AB250 pH meter. This meter was calibrated before testing with 4, 7, and 10 pH standards. Total P and TK were determined by ashing samples for at least 5 h (not to exceed 16 h) at 550 °C in a muffle furnace followed by extraction with 6M HCl and analyzed using an inductively coupled plasma-optical emission spectrometer (5110 ICP-OES, Agilent Technologies Inc., Santa Clara, CA, USA) (EPA method 200.7). Plant-available P and K in soil were measured using Mehlich-3 extraction. Two grams of soil was weighed and transferred into a 50 mL extraction bottle and 20 mL of Mehlich-3

extracting solution was dispensed into each extracting bottle with a pipette. Samples were shaken for 5 min on a reciprocating shaker and then filtered through a Whatman No. 42 filter paper. The filtrates were analyzed for M3P and M3K using 5110 ICP-OES, Agilent Technologies Inc., CA. Root, stem, panicle, and grain were analyzed for TP and TK by ashing 0.4 g dry tissue sample for 5 h at 550 °C in a muffle furnace followed by extraction with 6M HCl and analyzed using 5110 ICP-OES, Agilent Technologies Inc., Santa Clara, CA, USA. The spectrometer was first calibrated with standard solutions and a blank before measuring the sample.

**Table 1.** Brief description of the 14 rice genotypes used in this study (adapted from Hardke et al. 2019) [13].

| Genotype | Year Released/State | Description |
|---|---|---|
| Cheniere | 2003/Louisiana | A short season, semi-dwarf long-grain variety with good yield potential and milling quality comparable to Cypress. Susceptible to sheath blight and blast. |
| CL-151 | 2007/Louisiana | A mid-season, semi-dwarf long-grain Clearfield variety similar to Cocodrie with good yield potential. It is very susceptible to blast and straighthead, and susceptible to lodging and sheath blight. |
| CL-153 | 2016/Louisiana | A mid-season, semi-dwarf long-grain Clearfield variety similar to CL151 with good yield potential. Susceptible to sheath blight, kernel smut, and false smut. Moderately susceptible to blast. |
| CL-271 | 2011/Louisiana | Clearfield, medium grain variety with excellent grain quality. It is moderately resistant to blast and susceptible to sheath blight. |
| CL-272 | 2016/Louisiana | A mid-season, medium-grain Clearfield variety. High tolerance to Newpath herbicide. Very susceptible to bacterial panicle blight. Susceptible to sheath blight and blast. |
| Diamond | 2016/Arkansas | A mid-season, long-grain variety with excellent yield potential and good milling quality. Very good straw strength. Susceptible to blast and sheath blight, moderately susceptible to bacterial panicle blight. Very susceptible to false smut. |
| Dixiebelle | 1996/Texas | Short-season *long-grain* with 'Newrex' quality; specialty rice used for canning and steam tables. |
| Jupiter | 2006/Louisiana | A mid-season, semi-dwarf, medium-grain variety with excellent yield potential and milling quality. It has a small grain size but has moderate resistance to bacterial panicle blight. |
| LaKast | 2014/Arkansas | A mid-season, long-grain variety with excellent yield potential and good milling quality. Susceptible to blast and sheath blight. |
| Rex | 2010/Mississippi | A short season, semi-dwarf long-grain variety with excellent yield potential and good milling quality. Very good straw strength but is susceptible to most diseases. |
| Sierra | 2005/Texas | An aromatic long-grain with the fragrance and cooking qualities of a basmati style rice. |
| Titan | 2016/Arkansas | A short season, medium-grain variety with excellent yield potential. Moderately susceptible to blast and bacterial panicle blight. It has a preferred large grain size |
| XL753 | 2011/RiceTec, Inc | A short season, long-grain hybrid with excellent yield potential. Resistant to blast, moderately susceptible to sheath blight and straighthead. |
| XL760 | 2014/RiceTec, Inc. | A short season, long-grain hybrid with good yield potential. |

**Table 2.** Monthly average maximum and minimum air temperature, humidity, rainfall, and solar radiation of the study site in 2017.

| Month | Air Temperature | | Humidity (%) | Rainfall (mm) | Solar Radiation $(MJm^{-2})$ |
|---|---|---|---|---|---|
| | Maximum (°C) | Minimum (°C) | | | |
| March | $26.0 \pm 3.0$ | $13.2 \pm 2.8$ | $76.8 \pm 5.6$ | 17.5 | $18.8 \pm 3.9$ |
| April | $28.1 \pm 2.4$ | $16.5 \pm 2.8$ | $78.6 \pm 8.8$ | 35.7 | $22.0 \pm 7.3$ |
| May | $30.8 \pm 1.9$ | $19.2 \pm 2.5$ | $81.6 \pm 6.2$ | 134.6 | $23.9 \pm 4.9$ |
| June | $30.5 \pm 2.2$ | $22.5 \pm 1.0$ | $90.8 \pm 4.4$ | 326.9 | $17.4 \pm 6.2$ |
| July | $32.0 \pm 1.5$ | $22.9 \pm 0.6$ | $89.3 \pm 3.6$ | 167.5 | $18.4 \pm 5.2$ |
| August | $32.3 \pm 2.0$ | $22.7 \pm 0.8$ | $88.1 \pm 4.6$ | 213.9 | $20.3 \pm 5.6$ |
| September | $31.5 \pm 1.6$ | $19.3 \pm 10.2$ | $88.6 \pm 3.7$ | 384.6 | $17.9 \pm 4.8$ |

## 3. Statistical Analysis

Analysis of variance was performed to determine the difference in tissue concentration of potassium and phosphorus and yield among 14 rice cultivars. The cultivar difference was analyzed with the GLIMMIX procedure in Statistical Analysis System (Version 9.3; SAS Inc., Cary, NC, USA). All tests were performed at a significance of 0.05. Since planting time was different, each location was analyzed separately. The relationship between soil extractable Mehlich-3 phosphorus and potassium concentration with tissue phosphorus and potassium concentration were tested using regression analysis.

## 4. Results

*4.1. Soil pH and Mehlich-3 Potassium and Phosphorus*

Soil pH and M3P were significantly different between locations (Table 3). At Site I, pH was relatively lower than at Site II, whereas M3P and M3K were relatively higher than at Site II (Figures 1 and 2). At both locations, cultivar had no significant effect on soil pH and extractable soil M3K and M3P (Table 4). At Site I, pH was lowest (6.8) for XL760 and highest (7.1) for Dixiebel (Figure 1c). At Site II, pH was lowest (6.9) for XL760 and highest (7.3) for Lakast (Figure 1d). At Site I, M3P concentration was lowest ($21.4$ mg kg$^{-1}$) for CL272 and highest ($36.4$ mg kg$^{-1}$) for Titan (Figure 1c). At Site II, M3P concentration was lowest ($11.2$ mg kg$^{-1}$) for Titan and highest ($20.5$ mg kg$^{-1}$) for Lakast (Figure 1d). At Site I, M3K concentration was lowest ($53.9$ mg kg$^{-1}$) for CL271 and the highest ($151.0$ mg kg$^{-1}$) for Dixiebel (Figure 2c). At Site II, M3K concentration was lowest ($64.8$ mg kg$^{-1}$) for CL272 and highest ($104.1$ mg kg$^{-1}$) for Sierra (Figure 2d). Titan M3P concentration was highest at Site I but lowest at Site II.

**Table 3.** Analysis of variance for the effect of location, variety, and interaction of location and variety on soil pH, soil extractable Mehlich-3 potassium (SM3K), soil extractable Mehlich-3 phosphorus (SM3P), total concentration of root potassium (RK), root phosphorus (RP), stem potassium (SK), stem phosphorus (SP) panicle potassium (PK), panicle phosphorus (PP), grain potassium (GK), grain phosphorus (GP), and yield of rice.

| Effect | Df | pH | SM3K | SM3P | RK | RP | SK | SP | PK | PP | GK | GP | Yield |
|---|---|---|---|---|---|---|---|---|---|---|---|---|---|
| Location | 1 | * | ns | * | * | * | * | * | * | * | * | * | * |
| Variety | 13 | ns | ns | ns | * | * | * | * | * | * | * | * | * |
| Location × variety | 13 | ns | ns | ns | * | * | * | * | * | * | * | * | ns |

* Indicated significant deference at $p < 0.05$. ns indicated no significant difference.

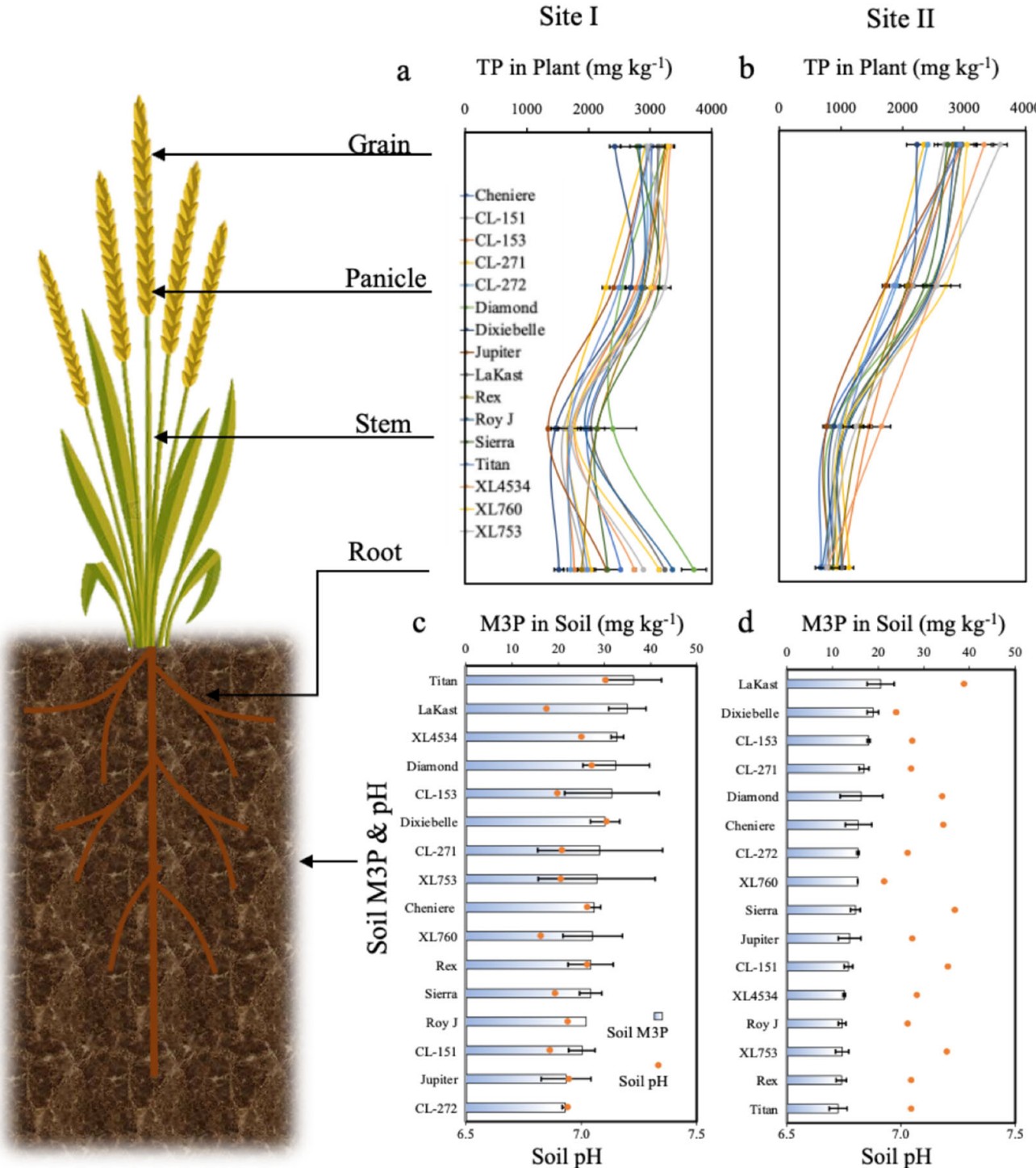

**Figure 1.** Total P concentration in rice grain, panicle, stem, and root at (**a**) Site I and (**b**) Site II. Soil concentration Mehlich-3 P and pH of Site I (**c**) and Site II (**d**). The bars represent standard deviations.

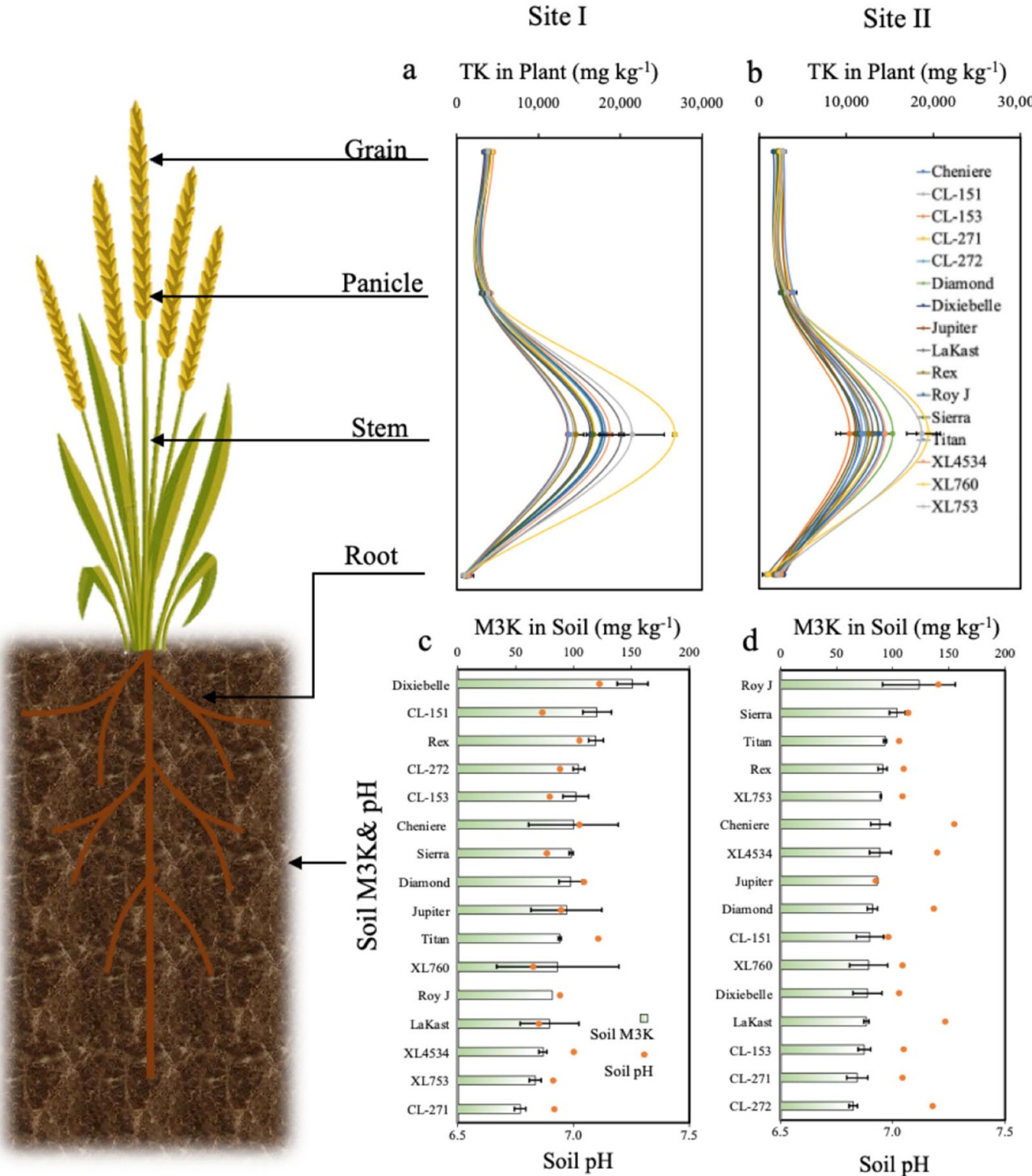

**Figure 2.** Total K concentration in rice grain, panicle, stem, and soil at (**a**) Site I and (**b**) Site II. Soil concentration Mehlich-3 K and pH of Site I (**c**) and Site II (**d**). The bars represent standard deviations.

**Table 4.** Analysis of variance for the effect of rice cultivars on soil pH, soil extractable Mehlich-3 potassium (SM3K), soil extractable Mehlich-3 phosphorus (SM3P), total concentration of root potassium (RK), root phosphorus (RP), stem potassium (SK), stem phosphorus (SP) panicle potassium (PK), panicle phosphorus (PP), grain potassium (GK), grain phosphorus (GP), and yield.

| Parameter | Site I | Site II |
|:---:|:---:|:---:|
| Soil pH | ns | ns |
| SM3K | ns | ns |
| SM3P | ns | ns |
| RK | ns | * |
| RP | * | ns |
| SK | * | * |
| SP | * | ns |
| PK | ns | * |
| PP | ns | * |
| GK | ns | * |
| GP | ns | * |
| Yield | ns | * |

*: significant difference at $p < 0.05$. ns: no significant difference.

### 4.2. Plant Total TP and TK

Root TP and stem TP concentration were significantly different among cultivars at Site I but not at Site II (Table 4). At Site I, panicle TP and grain TP concentration were not significantly different among cultivars, but they were significantly different at Site II. At Site I, root TP concentration was lowest (1520.3 mg kg$^{-1}$) for Dixiebel and highest (3707.0 mg kg$^{-1}$) for Diamond (Figure 1a). At Site II, root TP concentration was lowest (669.1 mg kg$^{-1}$) for Cheniere and highest (1135.6 mg kg$^{-1}$) for CL271 (Figure 1b). At Site I, stem TP concentration was lowest (1347.7 mg kg$^{-1}$) for Jupiter and highest (2392.7 mg kg$^{-1}$) for Diamond (Figure 1a). At Site II, stem TP concentration was lowest (770.2 mg kg$^{-1}$) for Jupiter and highest (1464.4 mg kg$^{-1}$) for CL152 (Figure 1b). Jupiter had the lowest stem TP at both sites. At Site I, panicle TP concentration was lowest (2280.8 mg kg$^{-1}$) for CL271 and highest (3234.2 mg kg$^{-1}$) for XL753 (Figure 1a). At Site II, panicle TP concentration was lowest (1730.4 mg kg$^{-1}$) for CL271 and highest (2705.0 mg kg$^{-1}$) for XL760 (Figure 1b). At both sites, panicle TP concentration was lowest for CL271, and XL753 had the highest at Site I and second highest at Site II. At Site I, grain TP was lowest (2426.3 mg kg$^{-1}$) for Dixiebel and highest (3289.8 mg kg$^{-1}$) for XL760 (Figure 1a). At Site II, grain TP concentration was lowest (2235.5 mg kg$^{-1}$) for Dixiebel and highest (3585.1 mg kg$^{-1}$) for XL753 (Figure 1b). Dixiebel had the lowest grain P concentration at both sites, whereas XL760 had the highest grain TP concentration at Site I and the second highest at Site II.

No significant difference in root TK was observed among cultivars at either location (Table 4). At both locations, stem TK was significantly different among cultivars. At Site I, panicle TK and grain TK were not significantly different among cultivars but significantly different at Site II. At Site I root K concentration was lowest (782.8 mg kg$^{-1}$) for CL272 and highest (1540.8 mg kg$^{-1}$) for Lakast (Figure 2a). At Site II, root TK concentration was lowest (898.4 mg kg$^{-1}$) for Lakast and highest (2800.4 mg kg$^{-1}$) for Dixiebel (Figure 2b). At Site I, stem TK concentration was lowest (13,550 mg kg$^{-1}$) for Jupiter and highest (26,635 mg kg$^{-1}$) for XL760 (Figure 2a). At Site II, stem TK concentration was lowest (10,387 mg kg$^{-1}$) for CL153 and highest (19,443 mg kg$^{-1}$) for XL760 (Figure 2b). At Site I, panicle TK concentration was lowest (3027 mg kg$^{-1}$) for Dixiebel and highest (3987 mg kg$^{-1}$) for Diamond (Figure 2a). At Site II, panicle TK concentration was lowest (2556 mg kg$^{-1}$) for Dixiebel and highest (3935 mg kg$^{-1}$) for Titan (Figure 2b). Dixiebel had the lowest panicle TK concentration at both sites. At Site I, grain K was lowest (3358 mg kg$^{-1}$) for Dixiebel and highest (4259 mg kg$^{-1}$) for XL760 (Figure 2a). At Site II, grain TK concentration was lowest (1636 mg kg$^{-1}$) for Dixiebel and highest (2809 mg kg$^{-1}$) for Jupiter (Figure 2b). Grain K concentration was lowest for Dixiebel at both sites. Root TP, stem TP, and panicle TP concentration increased with an increase in M3P (Figure 3a)

whereas root TK, stem TK, panicle TK, and grain TK concentration were not related to the increase in soil M3K (Figure 3b). None of the cultivars ranked consistently in the same top and bottom position for each measured parameter. Root TP, stem TK, and TP, panicle TP and TK, and grain TP and TK concentrations were higher at Site I compared to Site II, whereas root TK concentration was higher at Site II than at Site I.

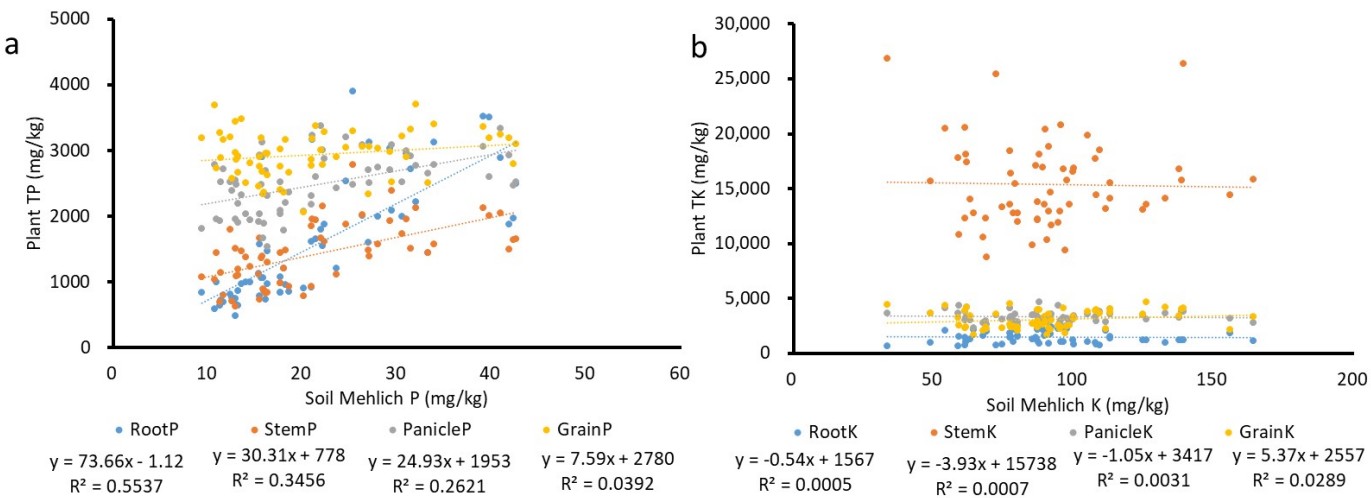

**Figure 3.** Relationship between (**a**) root TP, stem TP, panicle TP and grain TP and soil Mehlich-3 P, and (**b**) root TK, stem TK, panicle TK and grain TK and soil Mehlich-3 K.

*4.3. Yield*

No significant difference in yield was observed at Site I, whereas Site II had a significant difference in yield among cultivars (Table 4). At Site I, yield was lowest (3745.0 kg ha$^{-1}$) for CL272 and highest (7587.6 kg ha$^{-1}$) for CL271 (Figure 4). At Site II, yield was lowest (2627.3 kg ha$^{-1}$) for CL271 and highest (6406.9 kg ha$^{-1}$) for Cheniere (Figure 4). Site I had a higher yield than Site II. CL 271 had the highest yield at Site I but the lowest at Site II. XL760 had a higher yield at both locations with 7275.5 and 6387.4 kg ha$^{-1}$ at Site I and Site II, respectively.

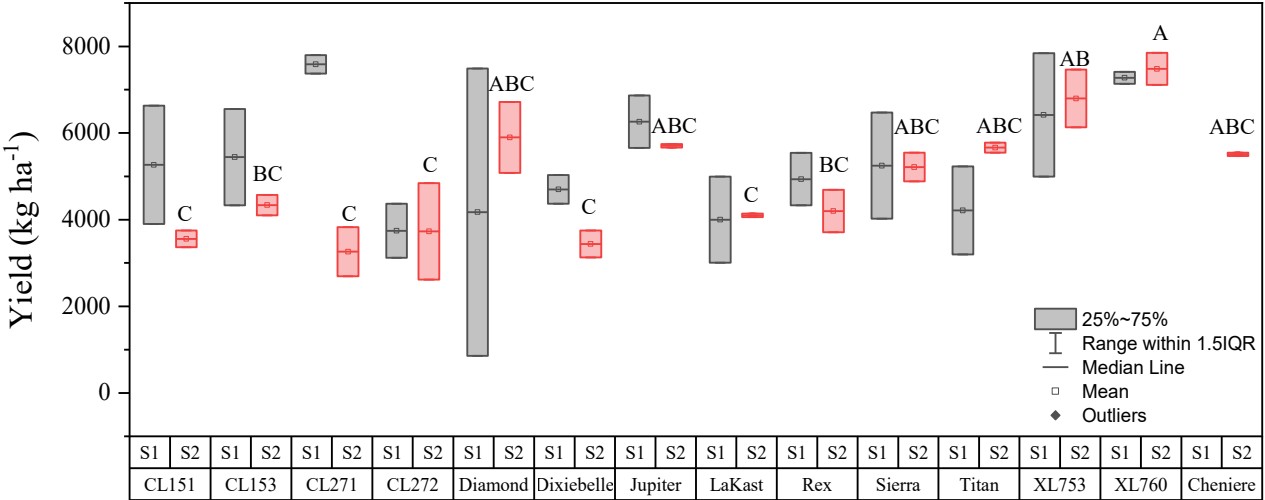

**Figure 4.** Rice yield (kg/ha) of different cultivars at Site I (S1) Site II (S2). Means sharing a common letter within each treatment at Site II (S2) are not significantly different at the *p* = 0.05 significance level. Note: No yield data were available for Cheniere cultivar at Site I.

## 5. Discussion

The lack of significant variations in soil pH or extractable soil M3K and M3P indicates that rice cultivars absorb the same amount of phosphorus and potassium from the soil. The results found in this study indicate narrow genotypic variation in nutrient uptake under histosol soil conditions. The lack of variability may be attributed to the high nutrient content in histosol. A previous study revealed significant genetic variability in nitrogen content under low soil fertility [14]. Nutrient concentrations depend on plant species or genotypes, age, and the plant part analyzed. The genetic differences in P and K absorption, as well as the differences in P and K levels in tissues, have been investigated previously [15,16]. Previous research found that more K was stored in the stem, whereas more P was stored in the panicle [15,16]. Since more K was stored in the stem, straw removal significantly affects the K-fertility status of the soil. Incorporating straw into the soil adds potassium and other nutrients to the soil. The extensive movement of carbon and other nutrients, including phosphorus (P), from vegetative organs to developing seeds occurs during the senescence stage in annual plants [17]. Between 6 and 15 days after flowering, phosphorus is quickly loaded into the rice cultivars' grains, while the physiological and genetic processes controlling P transfer into the grain are unknown. At maturity, P in grains can derive from two sources: exogenous P absorption from soil or endogenous P remobilization from the vegetative sections of the plant [17]. Because many soils are fundamentally deficient in accessible P, the storage of significant amounts of P in seeds gives emerging seedlings a competitive edge in natural environments [18].

The lack of variability in grain yield among cultivars might be due to the presence of high amounts of nutrient in the soil. Ref. [19] reported higher variability in the yield of rice cultivars with lower nutrient availability than with an optimal nutrient supply. This might be related to the fact that grain yield and growth are more dependent on nutrient absorption and use in low-fertile soils when nutrients are scarce [20]. This may be due to the different degrees of adaptability of different rice cultivars to low-fertile soil [14]. Previous research has reported that, regardless of soil fertility state, rice cultivar grain dry weight is strongly linked to shoot N, P, and K contents and accumulation rates [20].

## 6. Conclusions

The distribution and concentration of phosphorus and potassium in different parts of rice plants were evaluated. Phosphorus concentration was highest in the grain and lowest in the stem, whereas potassium was highest in the stem and lowest in the grain. Rice cultivars differed in yield, but not in nutrient concentration. Under the conditions encountered in this study, none of the cultivars ranked consistently in the same top and bottom position for each measured parameter at both locations. A high concentration of P and K in stem and panicle suggests that it can be incorporated as a potential soil amendment to offset the cost of fertilizer. The selection of cultivars should be based on other yield attributing characteristics, such as disease and insect resistance and milling quality.

**Author Contributions:** Conceptualization, J.H.B. and M.T.V.; methodology, J.H.B. and M.T.V.; formal analysis, N.R.A., Y.F. and A.R.; data curator, J.H.B., N.R.A.; writing-original draft preparation, J.H.B., N.R.A. and Y.F.; writing-review and editing, N.R.A., Y.F., A.R., J.H.B. and M.T.V.; visualization, J.H.B.; supervision, J.H.B. and M.T.V.; project administration, J.H.B.; funding acquisition, J.H.B. All authors have read and agreed to the published version of the manuscript.

**Funding:** This research was funded by Florida Rice Growers' Inc.

**Institutional Review Board Statement:** Not applicable.

**Informed Consent Statement:** Not applicable.

**Data Availability Statement:** All analyzed data have been reported here in the paper.

**Acknowledgments:** The authors would like to thank Salvador Galindo and Buazza Sakar for assistance in the field. We wish to thank the Florida Rice Growers' Inc. for funding this study; and NSF-STEPS Center CBET-2019435 for inspiring the work related to phosphorus sustainability.

**Conflicts of Interest:** The authors declare no conflict of interest.

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
