# Peer review of "From Ground to Grain: Tracing Phosphorus and Potassium in Flooded Rice Cultivar Grown on Histosols"

_agriculture, doi:10.3390/agriculture12081250_

Round 1

Reviewer 1 Report

Dear Authors,

I carefully read your manuscript and I mean that it is very interresting - both in scientific and practical aspects. I don't have substantive comments.  I see that you used appriopriate methods of experiment. The references are adequate, the descriptions of the results are also correct and interresting. I have just some editorial comments and suggestions:

line 63 - are you sure that cited Mehlich 3-P concentration is correct? You wrote: "123±170 mg kg-1 " - it suggest that in some conditions concentration can be below 0.

line 73 - in general the objective of your study is clear and well understood but I think that the fragment "...was to (i) track P, and K content..." could be extended. The tracking is just a part of your activity - I suggest the change this sentence. You can use the wrods "analyze", "compare" or other similar (with the meaning broader than tracking).

line 118 - please correct the title of 4.1 subsection - the word "Mehlich"  commes from the name of researcher - so it should be written with a capital letter (this comment also concerns the line 131).

Table 2 and 3 - in Table 2 you used just the shortcuts (in first line) while in Table 3 the full names were used (in the left column). I suggest to use the shortcuts in both tables (with the explanations in the titles of the tables). Moreover, please explain the symbols " * "and "ns" (I suppose that " * " means significant influence ans "ns" means "no signicance" but it should be written near the tables.

line 148 - probably there is a double space between the sentences in this line.

Figures 1 and 2 have very interesting form but I suggest to write the information about the error bars (what do they mean? Standard deviations?)

Results description - in some cases you use one decimal place in concentrations values while in other cases there is no decimal places - I suggest to use one form of values reporting.

Figure 3 - the letters on this figure could be slightly freater.

Reference 13 (in references list) - please use one date of publication of this reference. In the text you write year 2018 (line 84) while in references there is year 2019 as publication date.

Sincerely

Author Response

Comment1: line 63 - are you sure that cited Mehlich 3-P concentration is correct? You wrote: "123±170 mg kg-1 " - it suggest that in some conditions concentration can be below 0.

Response1: Due to the high variability in soil M3P data the standard deviation is greater than the mean, this does not mean that the concentration is below 0.

Comment2: line 73 - in general the objective of your study is clear and well understood but I think that the fragment "...was to (i) track P, and K content..." could be extended. The tracking is just a part of your activity - I suggest the change this sentence. You can use the wrods "analyze", "compare" or other similar (with the meaning broader than tracking).

Response2: The word “track” is replaced by “analyze and “compare”.

Comment3: line 118 - please correct the title of 4.1 subsection - the word "Mehlich"  commes from the name of researcher - so it should be written with a capital letter (this comment also concerns the line 131).

Response3: The word “Mehlich” is written with a capital letter.

Comment4: Table 2 and 3 - in Table 2 you used just the shortcuts (in first line) while in Table 3 the full names were used (in the left column). I suggest to use the shortcuts in both tables (with the explanations in the titles of the tables). Moreover, please explain the symbols " * "and "ns" (I suppose that " * " means significant influence ans "ns" means "no signicance" but it should be written near the tables.

Response4: Abbreviations are used in table 3. The symbols are explained at the bottom of table.

Comment5: line 148 - probably there is a double space between the sentences in this line.

Response5: In line 148- double space before “In Site I…” is deleted.

Comment6: Figures 1 and 2 have very interesting form but I suggest to write the information about the error bars (what do they mean? Standard deviations?)

Response6: The information about error bars is added. They represent standard deviations.

Comment7: Results description - in some cases you use one decimal place in concentrations values while in other cases there is no decimal places - I suggest to use one form of values reporting.

Response7: Results are reported to only one decimal point.

Comment8: Figure 3 - the letters on this figure could be slightly greater.

Response8: Font size of figure 3 is increased.

Comment9: Reference 13 (in references list) - please use one date of publication of this reference. In the text you write year 2018 (line 84) while in references there is year 2019 as publication date.

Response9: Publication date in the text (line 84) is changed to 2019.

Reviewer 2 Report

The author may add weather information for both locations.

The author may give the novelty of the data presented.

The manuscript is well written and well documented, the results presented may have merit, but do not explicitly enhance mechanistic understanding of plant-soil interactions, uptake, yields, etc. Such information is already available and well documented. 

Author Response

Comment10: The author may add weather information for both locations.

Response10: Weather information is added in table 2.

Comment11: The author may give the novelty of the data presented.

Response11: The novelty of data is presented in line 70-74. “This study provides an insight into the behavior of rice uptake of P and K from the soils to the gain without adding P or K in the form of fertilizer being applied during rice cultivation. The baseline data from this study will also be used to identify high yielding cultivars for histosols with reduced nutrient inputs.”

Comment12: The manuscript is well written and well documented, the results presented may have merit, but do not explicitly enhance mechanistic understanding of plant-soil interactions, uptake, yields, etc. Such information is already available and well documented. 

Response12: Thank you for your comments and suggestions. Mechanism of nutrient uptake by plants and soil-plant interactions was beyond the scope of this study. Therefore, the mechanism and soil-plant interaction were not disused in this paper.

Round 2

Reviewer 2 Report

Data present is sufficient to meet the objectives of the study, though article may be accepted in present form